# Analysis of Urinary Incontinence in the Neurogenic Bladder and Its Relationship with the Satisfaction and Lifestyle of People with SCI

**DOI:** 10.3390/healthcare12151501

**Published:** 2024-07-29

**Authors:** Lorena Gomes Neves Videira, Letícia Noelle Corbo, Marla Andreia Garcia de Avila, Giovana Pelosi Martins, Soraia Dornelles Schoeller, Christoph Kappler, Fabiana Faleiros

**Affiliations:** 1Ribeirão Preto College of Nursing, University of São Paulo, Ribeirao Preto 14040-902, Brazil; logomesvideira@gmail.com (L.G.N.V.); leticia.corbo@usp.br (L.N.C.); giovanapmartins@alumni.usp.br (G.P.M.); 2Faculty of Medicine of Botucatu, São Paulo State University, Botucatu 18618-687, Brazil; marla.avila@unesp.br; 3Federal University of Santa Catarina, Florianópolis 88040-900, Brazil; soraia.dornelles@ufsc.br; 4Faculty of Rehabilitation Sciences, University of Dortmund, 44227 Dortmund, Germany; christoph.kaeppler@tu-dortmund.de

**Keywords:** rehabilitation nursing, patient safety, chronic disease, patient safety, urinary incontinence, urinary bladder, neurogenic

## Abstract

One of the most common complications of neurogenic bladder secondary to spinal cord injury (SCI) is urinary incontinence, which is possibly related to bladder-emptying methods and changes in quality of life. This study aimed to identify the occurrence of this complication in adults with SCI and analyze its relationship with bladder-emptying methods, satisfaction, and lifestyle. This is a quantitative, exploratory, and cross-sectional study. The variables were collected using the Bowel and Bladder Treatment Index during a telephone interview with 290 participants from February to November 2021. According to the results, 70% of the participants were male and 74.1% performed clean intermediate catheterization (CIC) as the main bladder-emptying method. Moreover, 55.6% were considered incontinent in the last year. Emptying by normal urination and bladder reflex triggering had a statistically significant relationship with urinary incontinence. A statistical association was observed between all the variables of satisfaction and lifestyle with urinary incontinence. Although CIC reduced urine leakage, a considerable number of participants still presented with frequent urine leakage. Urinary incontinence had a negative impact on satisfaction with the bladder-emptying method, effectiveness of bladder management, quality of life, and personal and social relationships.

## 1. Introduction

Neurogenic bladder secondary to spinal cord injury (SCI) leads to complications that are possibly associated with the pathology of bladder dysfunction itself or with bladder management, thus negatively affecting health and quality of life, like urinary incontinence (UI) [1,2].

Urinary incontinence, defined as involuntary urine leakage during the storage phase of the bladder [3,4], is considered the most unpleasant of these complications and may even negatively affect participation in society [3].

Adequate bladder management is an important element of the rehabilitation of SCI, and one of its key objectives is the acquisition of urinary continence [5]. Despite the impact of this complication on people with spinal cord injury, little has been done with respect to the psychosocial consequences of neurogenic bladder and their impact on quality of life [6,7].

Considering that SCI is the main cause of neurogenic bladder, the number of people in the world living with SCI is not sufficiently clear as the number of publications remains low and most epidemiological studies on the incidence and prevalence of SCI come from developed countries [8,9].

In the USA, there is a national database that enables access to more precise numbers [10,11]. In Brazil, there is no national database, but Pelosi et al. (2021), in a large study carried out in one of the most important Brazilian neurological rehabilitation reference centers, found a prevalence of 94.65% of neurogenic bladder, which is somewhat higher than that described in some international studies that have reported between 70% and 84% [12,13,14]. In Brazil, a study evaluated the effects of urinary incontinence on the quality of life of people with spinal cord injury and demonstrated that urinary incontinence compromises QoL, especially in terms of social life, sexuality, and affectivity [15]. Another study conducted in Denmark evaluated bladder-emptying methods and urinary incontinence in individuals with long-term spinal cord injury. One of the results found in this study showed that a greater number of participants using clean intermittent catheterization (CIC) reported episodes of urinary incontinence than participants who used other bladder-emptying methods [3].

Thus, considering that urinary incontinence may be related to the bladder-emptying method and impact on quality of life, this study aimed to analyze urinary incontinence due to a neurologic bladder and its relationship with bladder management, satisfaction, and lifestyle in people with spinal cord injury.

## 2. Materials and Methods

This is a quantitative study with an exploratory, analytical, and cross-sectional design conducted from February to November 2021 at the Ribeirão Preto School of Nursing, Universidade de São Paulo. Since the records were obtained from a registry of people who voluntarily accepted to participate in research in this area, the scope of this study was nationwide.

The sample included adults, aged 18 years or older, with spinal cord injury who enrolled as volunteers to participate in research at the Center for Research and Care in Neuropsychomotor Rehabilitation (NeuroRehab). NeuroRehab, established in 2014, develops studies, technologies, and extension activities in rehabilitation to maximize the autonomy and participation of people with disabilities in society. For this purpose, NeuroRehab has a database of people who enrolled as volunteers to participate in research on spinal cord injury [16].

At the beginning of the survey, the registry had around 1200 subscribers. Those who did not indicate a mobile phone number or a landline, or were under 18 years old, were excluded. The sample size was calculated according to a population equal to 1200 and a *p*-value of 0.5. According to the level of significance, the number of participants necessary to achieve the desired margin of error was 290. For the sample calculation, the methodology proposed by Bolfarine and Bussab on proportions for finite situations was used [17]. Of the proportions for finite populations, *p* is the proportion of interest coming from the research instrument.

To enable the calculation of the sample size for the different variables with the specified levels of significance and margin of error, a *p* of 50% was used since the sample size obtained under this assumption was the maximum sufficient for any possible result that may occur.

Data were collected using two instruments. Data collected to characterize the sociodemographic and clinical profiles of the participants were based on ISCoS (International Spinal Cord Society) datasets [18]. The following variables were included: sex (male and female), date of birth, age, cause, date, level, and classification of spinal cord injury. The second instrument was the Bowel and Bladder Treatment Index (BBTI) developed by Tate et al. and validated and translated into Brazilian Portuguese by Braga [19,20]. The BBTI is a self-report instrument for patients with spinal cord injury. It was developed according to standards of the ISCoS and American Spinal Cord Injury Association to evaluate bowel and bladder management methods and complications, as well as the impact on quality of life, satisfaction, and lifestyle for use in patients with spinal cord injury during clinical face-to-face or telephone interviews.

The BBTI contains 60 items and is divided into sections on the intestine and bladder that address management methods, complications, personal satisfaction, and impact on life [18].

For this study, the section that evaluates the bladder domain was used to collect data on the impact of bladder function on quality of life, the bladder-emptying method, complications of neurologic bladder, satisfaction with bladder management, and its effect on lifestyle.

For each bladder-emptying method, participants can indicate whether it is the main method they use to manage their bladder or if it is used as a supplementary, and these answers were divided into the main or complementary method groups. Up to two main methods and more than two complementary supplementary methods may be indicated. The methods are as follows:-Normal urination: voluntary onset of urination without reflex, bladder compression, or stimulation;-Voluntary bladder reflex stimulation: tapping the bladder area and stretching the body to facilitate drainage;-Involuntary bladder reflex stimulation: incontinent in diapers or in collector and no perception of urination;-Bladder compression (straining): abdominal straining and Valsalva maneuver;-External compression: Credé maneuver and manual pressure in the suprapubic region;-Intermittent bladder catheterization: periodically inserting a catheter through the urethra into the bladder to allow urine to drain. The technique can be performed by the individual with SCI, known as self-catheterization, or by third parties, known as assisted catheterization;-Indwelling bladder catheter: a long-term catheter that remains inside the bladder. It can be inserted through the urethra, known as an indwelling transurethral catheter, or surgically inserted through the abdominal wall, known as an indwelling suprapubic catheter;-Non-continent urinary diversion/ostomy-stoma/diversion of urine through an opening created in the abdomen and Mitrofanoff.

The urinary complications item section contains questions such as the following asking about UI in the last 12 months: Have you had any involuntary urine leakages during the last year? (daily, not every day but at least once per week, not every week but at least once per month, less than once per month, or never).

To better evaluate the urinary incontinence variable, the participants were divided into 2 groups. Those who responded as having daily leakage at least once a week or at least once a month were included in the incontinence group, and those who reported leakages at less than once a month or no leakages were included in the continence group.

Satisfaction with the adopted bladder management method and the effects of bladder management on lifestyle in the continence or incontinence group were measured with questions on satisfaction with the bladder management routine (very dissatisfied, dissatisfied, satisfied, or very Satisfied); the causes of dissatisfaction with the bladder management routine; lifestyle changes due to urine leakage (none, less than once a month, at least once a month, at least once a week, or daily); effects of bladder problems on quality of life (no effect, little effect, some effect, or severe effect); and whether bladder management prevents the respondent from working outside the home, from performing their usual activities, prevents the respondent from leaving home, causes a problem for the respondent, or whether bladder management interferes with the respondent’s social life (not at all, a little, or very much).

For the descriptive statistical analysis of the categorical variables (qualitative variables), absolute and relative frequencies were used. In the description of the numerical variables (quantitative variables), measures of position, central tendency, and dispersion were used. To describe the numerical items of the satisfaction and lifestyle instrument, as well as to measure central tendency and dispersion, the bootstrap percentile interval of 95% confidence was used. To associate the variables of bladder management, UI, satisfaction, and lifestyle, the Chi-square test, the simulated Chi-square test, and Fisher’s exact test were used in cases of cross-tabulation. When all pairs of characteristics had values greater than 5, the Chi-square test was used. When at least one pair of characteristics obtained a value less than or equal to five and the two variables tested had two levels, Fisher’s exact test was used. When at least one pair of characteristics obtained a value less than or equal to five and at least one of the variables tested had three levels or more, the simulated Chi-square test was used. The Mann–Whitney test was used to verify the association between two-level categorical variables and numerical variables.

In the variable efficacy of routine bladder management, a Likert scale from 0 (very ineffective routine) to 10 (very effective routine) was used.

A significance level of <0.05 was adopted in all analyses. The software R (version 3.6.1) was used in the analyses.

## 3. Results

The sample consisted of 290 participants: 70% male and 30% female. The current mean age of the participants was 41.02 years (SD = 10.43), with a minimum age of 18 and a maximum of 74 years.

The mean time of SCI was 139.04 months (SD = 106.24) or approximately 12 years. The major causes of SCI were traumatic (79.30%), including traffic accidents (44.1%), automobile and motorcycle accidents, accidents with firearms and knives (14.50%), falls (10.00%), and diving in shallow water (9.00%). Non-traumatic causes totaled 21.70% and included congenital diseases (0.7%), myelopathies (15.50%), issues related to medical–surgical procedures (3.80%), or other non-traumatic causes (1.7%).

Regarding the classification of spinal cord injury, 67.70% reported having paraplegia and 34.40% reported having quadriplegia, with a greater occurrence at the thoracic level (62.80%), followed by cervical level (32.30%) and lumbar level (4.90%). Participants with incomplete spinal cord injury totaled 44.13%, and complete spinal cord injury totaled 35.60%. Moreover, 20.30% of the participants did not know whether the lesion was complete or incomplete.

Table 1 shows a descriptive analysis of bladder management methods. Clean intermittent catheterization was the most widely used emptying method (74.10%), with 57.20% of the participants performing self-catheterization and 16.90% performing catheterization assisted by third parties. Only one participant reported having a non-continent urinary diversion/ostomy. The methods of anterior sacral root stimulation and Mitrofanoff were not mentioned.

Regarding bladder emptying, almost half of the participants (43.70%) emptied their bladder 5 times a day, 20.50% 4 times a day, 20.50% 6 times a day, 10.80% 7 times or more, and 4.50 1 to 3 times a day. The mean daily bladder emptying according to a method was higher for bladder compression (m = 7.40 times/day), followed by normal urination (m = 6.72 times/day) and CIC (m = 5.05 times/day).

Regarding the frequency of involuntary urine leakage in the last year, 37.00% answered they had leakage daily, 13.10% at least once a week, and 5.50% at least once a month, that is, 55.60% were considered incontinent. In the continence group, 29.10% never had leakage and 15.20% had a leakage less than once a month. In this study, 51.00% of the participants used a urinary device, such as a diaper device (65.50%), followed by an external collector (33.80%) and an ostomy bag (0.70%).

When analyzing urinary incontinence by bladder-emptying method, 86.70% of the participants who performed bladder reflex triggering and 59.10% of the participants who performed clean intermittent catheterization were incontinent. Emptying by normal urination and bladder reflex triggering had a statistically significant relationship with UI.

In terms of the odds ratio (OR) of each management method, the chance of continence increased an average of 2.23 times for the participants who performed normal urination (Table 2).

Considering that the vast majority of the sample performed clean intermittent catheterization as a bladder-emptying method, a descriptive analysis of the urine leakage frequency in this group of participants was performed (Figure 1). In this analysis, 62.30% presented with some frequency of urine leakage (daily, at least once a week, less than once a month, or at least once a month), of which 37.70% had daily leakage.

### Satisfaction, Lifestyle, and the Relationship with UI

When the participants were asked about their level of satisfaction with their bladder-emptying routine, 58.60% in total either responded they were satisfied or very satisfied (Figure 2).

Although a smaller percentage of the sample (41.40%) stated they were dissatisfied or very dissatisfied with the bladder-emptying routine, when asked about the reason for their dissatisfaction, the second main cause was related to urinary incontinence (41.70%) (Table 3).

The participants were asked about the effectiveness of their bladder-emptying routine on a scale of 0 to 10. The mean was calculated with a bootstrap confidence interval and an average score of 7.75 (SD = 2.12). A statistical significance was observed for the effectiveness of the bladder-emptying routine with UI (*p* < 0.001, Mann–Whitney test) and satisfaction with the bladder-emptying routine (*p* < 0.001, Chi-square test).

Regarding the possible changes in lifestyle in the last year due to urine leakage in the participants with UI, 64.00% stated changes with some level of frequency and only 36.00% reported no changes to lifestyle due to leakage episodes (Table 4).

Statistical analyses were performed relating urinary incontinence to the categorical variables of satisfaction and lifestyle. An association was observed with the effect of bladder problems on life (*p* = 0.008, simulated Chi-square test); bladder management as an obstacle to working outside the home or doing routine activities (*p* < 0.001, Chi-square test); impaired personal relationships (*p* = 0.001, Chi-square test); and impaired social life (*p*-value = 0.002, Chi-square test) (Table 5).

## 4. Discussion

This study analyzed urinary incontinence and its relationship with bladder-emptying methods, personal satisfaction, and lifestyle in people with spinal cord injury. In this regard, the prevalent bladder-emptying method among participants was CIC (74.10%). Clean intermittent catheterization is recommended as the first-choice treatment for neurogenic bladder, and it is mainly indicated to protect the upper urinary tract, prevent and control urinary incontinence, and promote continence [5,19,20,21,22]. The predominance in this study of participants who performed clean intermittent catheterization demonstrates that most urologists indicate this method as the gold standard in the treatment for neurogenic bladder patients who exhibit urinary retention, an inability to adequately drain their bladders, or have larger post-void residual volumes (typically > 200 mL) [23].

The mean number of bladder emptying instances reported by the participants was 5.4 times a day. However, the mean was higher among participants who performed bladder compression (7.4 times a day). The natural emptying frequency for a typical bladder is four to six times a day depending on fluid intake and bladder volume. This frequency is also indicated in patients who perform clean intermittent catheterization according to urological guidelines [5]. Therefore, the result is consistent with the mean found in this study for emptying by CIC (5.04). Regarding bladder compression, the participants were expected to perform emptying more times a day because bladder compression does not guarantee complete emptying.

This study revealed that 55.6% of the participants had episodes of incontinence in the last year (2020). Of these participants, 37% had daily leakage and 51% used some device for incontinence, especially diapers. Urinary incontinence was also one of the main causes of dissatisfaction with routine bladder management (41.7%). Previous studies corroborate the findings of this study regarding urinary incontinence in people with SCI [6,23,24,25]. A study of the translation, localization, and validation of datasets of the lower urinary tract for people with spinal cord injury showed similar results in the use of urinary devices, with diapers as the most frequently used device [2].

When comparing each bladder management method, bladder reflex triggering had the highest percentage of individuals with urinary incontinence (86.7%), followed by clean intermittent catheterization (59.1%) and bladder compression (55.5%), which corroborates the results of previous studies [2,8,17]. Only emptying by normal urination showed a statistically significant association with urinary continence and bladder reflex triggering with UI.

The relationship between emptying by normal urination and urinary continence is expected since participants may be resuming normal bladder function that involves the coordinated action of the pelvic floor, detrusor muscle, and urethral sphincter, resulting from the coordination between the multiple regulatory centers in the brain, spinal cord, and peripheral nerves [26]. In contrast, the association of bladder reflex triggering with UI is expected because this method involves maneuvers performed by people with SCI or by a caregiver to voluntarily or involuntarily stimulate the detrusor reflex [2], which is often without perception of urine outflow and, thus, without continence control.

In this study, no statistically significant relationship was observed between clean intermittent catheterization and urinary incontinence, although it was the second most widely used method among the participants with incontinence (59.1%). A description of the frequency of urinary incontinence in lean intermittent catheterization users revealed that 37.7% had a urine leakage daily, 16.0% had a leakage less than once a month, and 24.1% never had a leakage. Studies in the literature approximate the percentage of UI in the users of CIC17.

One of the objectives of clean intermittent catheterization is to improve urinary incontinence. However, the person often acquires partial continence, as demonstrated in this study, resulting in a decrease in leakage frequency, which improves QoL. In contrast, the entire rehabilitation team should address both the expectations of patients before initiating CIC to minimize the chances of abandoning the technique [27,28] and the health education of this population. Insufficient health literacy can affect the patient’s ability to take care of themselves and has a direct negative effect on their health, thus increasing the risk of complications [29,30].

In the discussion about urinary incontinence acquisition after clean intermittent catheterization, the use of drug therapy for the bladder should be considered as adjuvant in the overall treatment. These drugs assist in the acquisition of continence by decreasing detrusor hyperactivity and increasing bladder capacity or sphincter insufficiency [20,31], which reduces leakage.

Satisfaction with the bladder-emptying routine and the effectiveness of the routine were related to urinary continence in this study. One of the objectives of bladder management is the acquisition of urinary continence. Therefore, participants are expected to feel more satisfied with their routine since urinary continence affects the QoL of people with spinal cord injury [25,26,32].

When asked about possible changes in lifestyle as a result of urine leakage, 64.0% stated they changed their lifestyle with some type of frequency, and 28.0% among these changed their lifestyle daily. The results are similar to the findings of a national study that evaluated the impact of UI on the QoL of people with SCI [15]. Moreover, another study that compared health-related QoL, daily activities, and the use of health resources between patients with continence and incontinence found that the daily activities of the participants with incontinence were more greatly affected than the daily activities of the patients with continence [7].

When relating UI with the variables of satisfaction and lifestyle, it was found that bladder problems had a greater effect on QoL, difficulty working outside the home, performing routine activities, and impaired personal relationships and social life.

Previous studies corroborate these findings concerning the negative effect of UI on the QoL of people with SCI [25,26] and impaired social, personal, and affective relationships [15]. The uncertainty of patients regarding the risk of urine leakage causes social isolation to avoid embarrassment and concerns about UI, as shown in a qualitative study that describes the experiences of bladder and intestinal dysfunction in people with SCI. According to that study, these dysfunctions have a profound effect on relationships and social activities, especially outside the home, and they are also related to the risk of incontinence [33,34,35].

These data reinforce the need and importance for UI control in people with SCI. In this regard, the need to change the entire bladder-emptying routine greatly affects the lifestyle of these people who, when choosing a management method, do not expect further concerns with UI, which is also considered an important factor for resuming their social, work, and affective lives.

Based on the above considerations, nursing professionals should discuss ways to maximize urinary continence and minimize the resulting damage. The rehabilitation nurse should seek care alternatives to ensure people with UI can live to their fullest, with social participation and QoL.

### Study Limitations

This study has some limitations. First, the data collected for this study were based on the perception of the participants regarding bladder care. Thus, at the time of the interview, some participants may not remember some information about forms of bladder care and complications. However, memory bias is often present in cross-sectional studies.

Second, the participants may have encountered some difficulties interpreting the questions since the interview was conducted over the telephone. However, to minimize any difficulties, the interviewers were trained to conduct the interviews in a standard manner that was clear and easy to understand. Furthermore, due to social distancing during the COVID-19 pandemic, data collection by telephone ensured the participation of people from all over Brazil with no risk to participants, researchers, and interviewers.

Given the cross-sectional design of this study, changes in the frequency of urinary leakage after initiating CIC were not evaluated. Since one of the objectives of CIC is to reduce urine leakage, longitudinal studies are suggested.

Data on the use of bladder medications would provide valuable insight into the large number of participants with UI, especially among those who perform CIC. Drug therapy associated with the bladder management method reduces the chance of UI in terms of decreasing detrusor hyperactivity, increasing bladder capacity, or reducing sphincter insufficiency. It will be suggested to the authors of the instrument to include these data.

## 5. Conclusions

This study showed that bladder management is not yet a fully resolved issue when evaluating the relationship with UI. Despite advances in urinary tract protection and the consequent decrease in mortality, UI still has a negative effect on the lives of people with SCI, even with the indicated emptying methods.

CIC was the predominant bladder-emptying method. Despite reducing urine leakage, a considerable number of participants still had frequent leakage, thus preventing the complete acquisition of urinary continence. Bladder emptying with bladder reflex triggering was associated with UI, while normal urination increased the chances of continence.

Satisfaction and lifestyle were strongly affected by UI due to the BN in the participants of this study. Moreover, UI negatively affected satisfaction with the bladder management routine, effectiveness of bladder management, QoL, and personal and social relationships.

Given the results obtained in this study, measures should be adopted to minimize the effect of UI on the lives of people with SCI and BN, including the longitudinal monitoring of bladder dysfunction and an individualized indication of a bladder-emptying method that considers the expectations of acquiring urinary continence. In addition, continuing education should be provided to rehabilitation professionals, especially nurses, in order to empower patients and manage the physiological and emotional consequences that may result from bladder dysfunction and its complications.

In this regard, future studies are needed to better understand these findings and improve the current methods of managing NB. These improvements include the development of new care technologies that increase the chances of acquiring urinary continence and, consequently, bettering the lives of people with SCI and BN.

## Figures and Tables

**Figure 1 healthcare-12-01501-f001:**
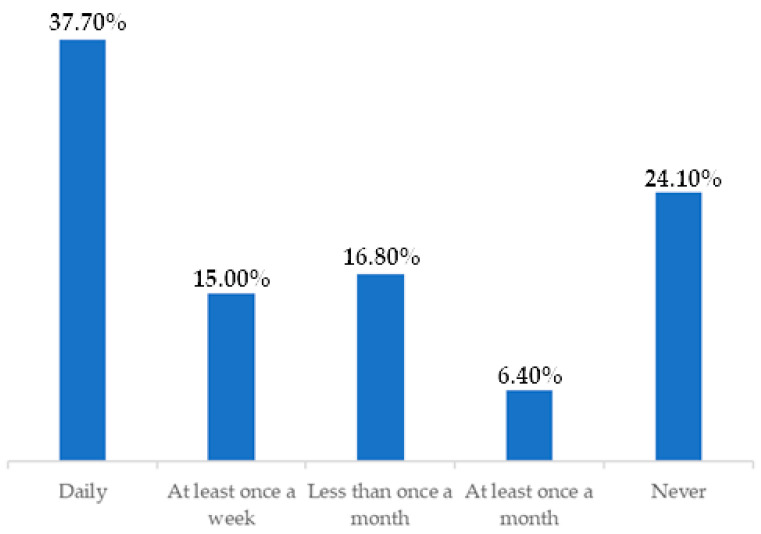
Frequency of urine leakage in the CIC management method. n = 220. Brazil, 2021.

**Figure 2 healthcare-12-01501-f002:**
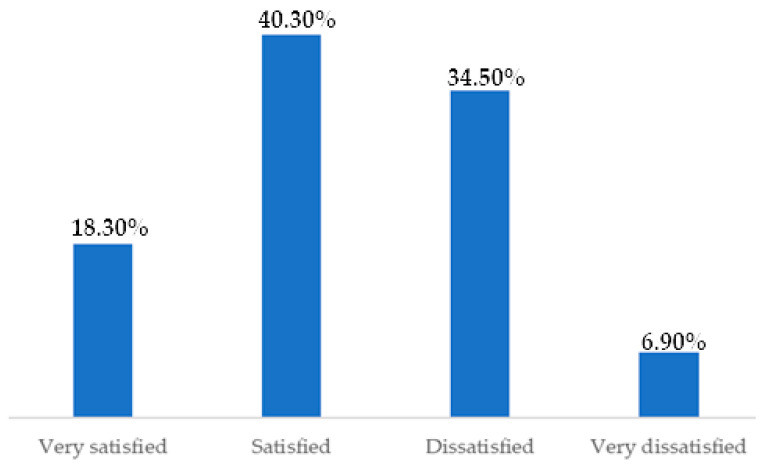
Sample distribution according to the level of satisfaction with bladder-emptying routine. n = 290. Brazil, 2021.

**Table 1 healthcare-12-01501-t001:** Sample distribution according to bladder-emptying method in the last 4 weeks (n = 290). Brazil, 2021. Source: Prepared by the authors.

Bladder-Emptying Method	Main Method		Complementary Method	
	N	%	N	%
Normal urination	32	11	2	0.7
Voluntary and involuntary bladder reflex triggering	14	5.1	2	0.7
Bladder compression through forced effort (abdominal pressure and Valsalva maneuver) and external compression (Credé maneuver and manual pressure in the suprapubic region)	16	4.5	11	3.4
Clean intermittent catheterization (self-catheterization and assisted catheterization)	214	74.1	7	2.4
Indwelling transurethral and suprapubic bladder catheterization	11	3.7	-	-
Non-continent urinary diversion/ostomy	1	0.3	-	-

**Table 2 healthcare-12-01501-t002:** Association between bladder management methods and continence. n = 290. Brazil, 2021.

Management Method\Incontinence	Yes		No		*p*-Value	OR	CI 95%
	N	%	N	%			
Normal urination	13	38.2	21	61.8	0.046 ^1^	2.23	[1.08; 4.77]
Bladder reflex triggering	13	86.7	2	13.3	0.015 ^2^	0.18	[0.03; 0.67]
Bladder compression	15	55.5	12	44.5	0.999 ^1^	1.01	[0.45; 2.23]
Intermittent catheterization	130	59.1	90	40.9	0.054 ^1^	0.56	[0.32; 0.97]

^1^ Chi-square test. ^2^ Fisher’s exact test.

**Table 3 healthcare-12-01501-t003:** Sample distribution according to the causes of dissatisfaction with a bladder-emptying routine (n = 120). Brazil, 2021.

Causes of Dissatisfaction with Bladder-Emptying Routine	N	%
Difficulties related to bladder-emptying method	59	49.1
Related to UI	50	41.7
Architectural barriers	19	15.8
Psychological difficulties	13	4.5
Related to urinary tract infection	12	4.1

**Table 4 healthcare-12-01501-t004:** Descriptive distribution of the responses on lifestyle changes due to urine leakage in participants with incontinence. n = 161. Brazil, 2021.

Lifestyle Changes Due to Urine Leakage	N	%
None	58	36.0
Less than once a month	22	13.7
At least once a month	17	10.6
At least once a week	19	11.8
Daily	45	28.0

**Table 5 healthcare-12-01501-t005:** Association between incontinence and the categorical variables of satisfaction and lifestyle. n = 290. Brazil, 2021.

Satisfaction and Lifestyle\Incontinence		Yes		No		*p*-Value
		N	%	N	%	
Effects of bladder problems on QoL	No effect	16	9.9%	24	18.8	0.008
	Little effect	37	23.0%	36	28.1	
	Some effect	33	20.5%	32	25.0	
	Severe effect	75	46.6%	36	28.1	
Bladder management prevents me from working outside the home or doing routine activities	Not at all	55	34.2%	77	60.2	<0.001
	Somewhat	63	39.1%	35	27.3	
	Very much	43	26.7%	16	12.5	
Bladder management negatively affects my personal relationships	Not at all	61	37.9%	76	59.4	0.001
	Somewhat	55	34.2%	26	20.3	
	Very much	45	28.0%	26	20.3	
Bladder management negatively affects my social life	Not at all	56	34.8%	71	55.5	0.002
	Somewhat	69	42.9%	39	30.5%	
	Very much	36	22.4%	18	14.1%	

## Data Availability

The data presented in this study are available on request from the corresponding author due to respecting the privacy of the patients.

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
