# Peer review of "Analysis of Urinary Incontinence in the Neurogenic Bladder and Its Relationship with the Satisfaction and Lifestyle of People with SCI"

_healthcare, 2024, doi:10.3390/healthcare12151501_

Round 1

Reviewer 1 Report

Comments and Suggestions for Authors

The topic of the study is interesting and relevant to the provision of healthcare and its users.

I leave some suggestions for improvement:

You must review the referencing in accordance with the journal's rules.

The introduction does not provide an adequate framework for the topic. There is a lack of data on the incidence and prevalence of the problems under study, their definition and diagnostic criteria (Neurogenic bladder, spinal cord injury, urinary tract infections...).

Excessive use of acronyms makes the article difficult to read; they should only be used in longer terms that are used frequently in the article.

When calculating the required sample size, the source of information for the same must be indicated.

you should improve the discussion with more recent sources, more than 80% of citations are more than 5 years old.

In general, the article needs reinforcement in terms of theoretical foundations, both in the framework and in the methodology and discussion.

Comments on the Quality of English Language

Minor editing of English language required

Reviewer 2 Report

Comments and Suggestions for Authors

Dear Authors,

Thank you for your interesting paper. To further improve your paper, I suggest that you work on the following:

1. Authors' affiliation - Kindly review the use of the articles "a" and "an"

   example in your paper: an master, an professor

2. The title is not consistent with the study aim stated in the abstract. Kindly clarify if you are looking at the management of UI or the prevalence of UI...

Your title indicates that you are looking at the relationship between the management of UI with satisfaction and lifestyle. In your abstract, you stated that you aimed to determine the occurrence of UI and its relationship with bladder emptying methods, satisfaction, and lifestyle. In the introduction (lines 63-64), you also mentioned that the study aimed to analyze UI due to NB and its relationship with bladder management, satisfaction, and lifestyle.

Please be consistent with the  aim and the terms used ( determine, analyze, bladder management, bladder emptying methods...)

3. Abstract- Kindly organize the presentation of the results following your stated aim of the study.

The last sentence of the abstract (Lines 35-36) is not clear. What do you mean in this sentence? What do you mean by negatively altered?

4. Introduction - 

Please state your thesis statement clearly and make sure to support it with evidence and examples. It was not well discussed and you were not able to explain well the gap you wanted to address with your study. Also kindly justify further why your study is significant and relevant.

5. Materials and Methods

I suggest for you to organize this section by categorizing the information with subsections like: Research Design, Setting, Population and sample (inclusion and exclusion criteria, sampling technique), Instrument, Data Collection, and Data Analysis.

In the instrument section, it will be nice for you to describe how you measured your variables (prevalence of UI, Methods of bladder management, satisfaction, and lifestyle).

Did you measure the QoL or the lifestyle? Your title and aim stated lifestyle. What is the role of QoL in your study?

Did you use the English language or the vernacular during your survey? If yes, include this information under instrument section. Describe also how you validated the translation. What was the result of the pilot test? Cronbach's alpha?

In the data analysis, specify which data were categorical and numerical instead of just saying the variables. Indicate also which specific variables were tested by  F-test, Chi square, and Mann Whitney instead of just saying categorical or numerical variables.

Kindly include your reference in the classification of your participants into incontinence and continence groups. Also, what is your reference for the likert scale 0-10 in terms of the efficacy of bladder management?

6. Results

For the age, include the age range (Minimum-Maximum) in addition to the mean and SD.

In the description of Table 1, explain what you mean by main method and complementary method.

I suggest you remove the description: "Source: Prepared by the author" across all tables/figures in the result section.

You can add subheadings for the results. example: Sociodemographic Profile, SCI Classification, Bladder emptying Profile, Associations

Other than the numerical notations, I suggest that you add * on the p values that indicate significance.

How come you only looked at the CIC method when you identified the frequency of urine leakage? Although you mentioned the rationale that the majority of the participants used IC, this does not justify excluding the rest in determining the prevalence of leakage among all the participants. In Table 2 and in your discussion, you showed that bladder reflex triggering has the highest percentage of UI.I suggest for this part to be reconsidered and include all the participants.

In general, I suggest that the discussion be improved in terms of the depth as it is limited in its current form.

There is also confusion between QoL and lifestyle.

What do you mean by architectural difficulties? (Table 3)

What do you mean by lifestyle changes? Why was it measured by frequency?(Table 5). What is the overall lifestyle? How do you define lifestyle?

The mean score should have been taken to describe the lifestyle.

7. Discussion

Line 227- What is your evidence for saying that CIC is the gold standard other than being the most used method

Lines 236-239- Where did you get these data? What year is last year? Where are these data in the results section?

8. Conclusion

What can you then conclude from the findings? The current conclusion is just a restatement of the findings with an addition of some recommendations. Moreover, affect is different from association or relationship.

May I request for a copy of the questionnaire used in this study?

Thank you and good luck.

Comments on the Quality of English Language

There is a need to read carefully the manuscript and consider editing some language/grammar .

Reviewer 3 Report

Comments and Suggestions for Authors

Dear Authors.

Thank you for the opportunity to review your work. 

Education about urinary incontinence is essential to empower patients to take an active role in their disease process, reducing the consequences that can result. After reviewing and analysing your work, I would like to make a couple of recommendations and suggestions. I would be grateful if you would take my comments into account as I believe that they can complete your manuscript.

- In the introduction, the significance and consequences of urinary incontinence secondary to neurogenic bladder, as well as its management, are perfectly expressed. However, perhaps what I am missing is an indication of the magnitude of the problem, specifically in Brazil, where the study was carried out, and how much of the population is affected.

- In the methodology, between lines 72-83, the inclusion and exclusion criteria are mentioned. I wonder if the exclusion criteria included people who did not have the physical and/or mental capacity to fill in the questionnaires.

- Also, in lines 88 and 89 the questionnaire is referenced with Tate et al. (2015) and Braga (2018). In these cases, the year can be omitted, and the reference can follow: e.g. Tate et al. (11)

- In this same part, it talks about the BBTI questionnaire and that it consists of 60 items. However, I think it should be included which or how many items refer to the bladder section (which is the one used for the study) and how these items are scored (Likert-type scale perhaps?). The bladder questionnaire could be added in a table or additional material for visualisation.

- In the figures and tables, it could be omitted when writing "Source: own elaboration". It is understood that the authors have prepared the tables and graphs. Also, in figures 1 and 2, it would be convenient to change the order of the frequencies: normally, the results are put from lowest to highest or vice versa, not starting with "daily", then "never", then "less than once a month", etc. It can be confusing to read and understand the graph.

- Finally, in line 180, when talking about graph 1, it refers to 62.30% of the participants having some urine leakage (frequency). However, I understand that daily was 37.7%, never 24.1% and sometime would be the remainder (38.20%).

Best regards

Round 2

Reviewer 1 Report

Comments and Suggestions for Authors

I consider that the article has quality for publication.

Author Response

Thank you for your comments.